Why sauropods had long necks; and why giraffes have short necks

Taylor Michael P. dino@miketaylor.org.uk 1
Wedel Mathew J. 2
1 Department of Earth Sciences , University of Bristol , Bristol , England
2 College of Osteopathic Medicine of the Pacific and College of Podiatric Medicine , Western University of Health Sciences , Pomona, California , USA
Hutchinson John
Electronic publication date: 2013 Feb 12
Publication date: 2013
Volume: 1
Electronic Location ID: e36
Received 2012 Dec 3; Accepted 2013 Jan 19
Copyright: © 2013 Taylor and Wedel
Copyright year: 2013
Copyright holder: Taylor and Wedel
License: This is an open access article distributed under the terms of the Creative Commons Attribution License, which permits unrestricted use, distribution, and reproduction in any medium, provided the original author and source are credited.
License URL: https://creativecommons.org/licenses/by/3.0/

Keywords: Sauropod, Giraffe, Dinosaur, Evolution, Cervical vertebra, Neck

Funding: No funding received.

==============================
The necks of the sauropod dinosaurs reached 15 m in length: six times longer than that of the world record giraffe and five times longer than those of all other terrestrial animals. Several anatomical features enabled this extreme elongation, including: absolutely large body size and quadrupedal stance providing a stable platform for a long neck; a small, light head that did not orally process food; cervical vertebrae that were both numerous and individually elongate; an efficient air-sac-based respiratory system; and distinctive cervical architecture. Relevant features of sauropod cervical vertebrae include: pneumatic chambers that enabled the bone to be positioned in a mechanically efficient way within the envelope; and muscular attachments of varying importance to the neural spines, epipophyses and cervical ribs. Other long-necked tetrapods lacked important features of sauropods, preventing the evolution of longer necks: for example, giraffes have relatively small torsos and large, heavy heads, share the usual mammalian constraint of only seven cervical vertebrae, and lack an air-sac system and pneumatic bones. Among non-sauropods, their saurischian relatives the theropod dinosaurs seem to have been best placed to evolve long necks, and indeed their necks probably surpassed those of giraffes. But 150 million years of evolution did not suffice for them to exceed a relatively modest 2.5 m.

Introduction

Neck elongation occurs in many extant clades and is also found in many extinct groups. Some modern birds and certain extinct tetrapods have necks that are relatively long (i.e. as a proportion of total body length). Although these are interesting modifications of the basic tetrapod body plan, here we are concerned with absolute neck length. This is of interest because of the great mechanical difficulties imposed by absolutely long necks, and the anatomical novelties that needed to evolve to make such necks possible.

The necks of the sauropod dinosaurs were by far the longest of any animal, six times longer than that of the world record giraffe and five times longer than those of all other terrestrial animals. In many long-necked animals, the legs are of a similar length and so the neck elongation can be explained as a simple consequence of the need to reach down to ground level – for example in order to drink. By contrast, the necks of all sauropods were longer than necessary to reach the ground – in most cases, many times longer.

We survey the evolutionary history of long necks in sauropods and other animals, and consider the factors that allowed sauropod necks to grow so long. We then examine the osteology of sauropod necks more closely, comparing their cervical anatomy with that of their nearest extant relatives, the birds and crocodilians, and discussing unusual features of sauropods’ cervical vertebrae. Finally we discuss which neck elongation features were absent in non-sauropods, and show why giraffes have such short necks.

Long Necks in Different Taxa

While they reach their zenith in sauropods, long necks have evolved repeatedly in several different groups of tetrapods. Long necks impose a high structural and metabolic cost, but provide evolutionary advantages including an increased browsing range (Cameron & du Toit, 2007) and the ability to graze a wide area without locomotion (Martin, 1987) and probably played some role in mate attraction (Simmons & Scheepers, 1996; Senter, 2006; Taylor et al., 2011). Here we survey the longest necked taxa in several groups of extant and extinct animals (Figs. 1 and 2).

Figure 1 Necks of long-necked non-sauropods, to scale.

The giraffe and Paraceratherium are the longest necked mammals; the ostrich is the longest necked extant bird; Therizinosaurus and Gigantoraptor are the largest representatives of two long-necked theropod clades; Arambourgiania is the longest necked pterosaur; and Tanystropheus has a uniquely long neck relative to torso length. Human head modified from Gray’s Anatomy (1918 edition, fig. 602). Giraffe modified from photograph by Kevin Ryder (CC BY, http://flic.kr/p/cRvCcQ). Ostrich modified from photograph by “kei51” (CC BY, http://flic.kr/p/cowoYW). Paraceratherium modified from Osborn (1923, figure 1). Therizinosaurus modified from Nothronychus reconstruction by Scott Hartman. Gigantoraptor modified from Heyuannia reconstruction by Scott Hartman. Arambourgiania modified from Zhejiangopterus reconstruction by Witton & Naish (2008, figure 1). Tanystropheus modified from reconstruction by David Peters. Alternating blue and pink bars are 1 m tall.

Figure 2 Full skeletal reconstructions of selected long-necked non-sauropods, to scale.

1, Paraceratherium. 2, Therizinosaurus. 3, Gigantoraptor. 4, Elasmosaurus. 5, Tanystropheus. Elasmosaurus modified from Cope (1870, plate II, figure 1). Other image sources as for Fig. 1. Scale bar = 2 m.

Note that most of the length estimates in this section are necessarily imprecise, being based on incomplete specimens and cross-scaling assumptions. They can be taken only as indicative, not as reliable figures.

Table 1 Neck statistics of some sauropods, chosen because of unusually long, short or complete necks.

Taxon	Neck length (m)	Cervical count	Longest centrum (cm)	Longest cervical rib (cm)	Maximum elongation index	
Mamenchisaurus hochuanensis	9.5	19	73  (C11)	210 (C14)	2.9  (C6)	
Mamenchisaurus sinocanadorum	12 est.	19?		 ≥ 410		
Brachytrachelopan mesai	1.1 est.	12?	10	 ≤ centrum	 ≤ 1	
Apatosaurus louisae	5.9	15	55  (C11)	39  (C11)	3.7  (C4)	
Diplodocus carnegii	6.5	15	64  (C14)	48  (C11)	4.9  (C7)	
Barosaurus lentus	8.5 est.	16?	87  (C14)	<centrum	5.4  (C8)	
Supersaurus vivianae	15.0 est.	15?	 ≥ 138	 ≤ centrum		
Giraffatitan brancai	8.5	13	100 (C10)	290 (C7)	5.4 (C5)	
Sauroposeidon proteles	11.5 est.	13?	125 (C8)	342 (C6)	6.1 (C6)	
Euhelopus zdanskyi	4.0	17	28  (C11)	72  (C14)	4.0 (C4)	

Extant animals

Among extant animals, adult bull giraffes can attain a neck of 2.4 m (Toon & Toon, 2003, p. 399), and no other extant animal exceeds half of this. The typical length of the neck of the ostrich is only 1.0 m (sum of vertebral lengths in Dzemski & Christian (2007), Table 1, plus 8% to allow for intervertebral cartilage – see Cobley, 2011, p. 16).

Extinct mammals

The largest terrestrial mammal of all time was the long-necked rhinoceratoid Paraceratherium Forster-Cooper, 1911 (= Baluchitherium Forster-Cooper, 1913, Indricotherium Borissiak, 1915). The length of its neck can be measured as 1.95 m from the skeletal reconstruction of Granger & Gregory (1936, figure 47). This length, however, is rather shorter than suggested by the text (pp. 10–20), in which lengths of 39, 39, 36, 29.6 and 18 cm are given for cervicals 1, 2, 4, 6 and 7, even though C2 and C7 are reported as of “size class III”. When the lengths of C2 and C7 are multiplied by 1.3 to give lengths of equivalent “size class I” bones (Granger & Gregory, 1936, p. 65), their lengths become 50.7 and 23.4 cm. The total length of the preserved cervicals would then be 178.7 cm even though C3 and C5, which were not recovered, are omitted. If these vertebrae are tentatively assigned lengths intermediate between those that preceded and succeeded them (i.e., 43.4 and 32.8 cm) then the total length of all seven centra is 254.9 cm, more than 30% longer than the illustrated length. At any rate, the material available suggests a total neck length in the 2–2.5 m range.

Theropods

In each of the successively more derived clades Ornithodira, Dinosauria, and Saurischia, the primitive state was an increasingly long neck (Gauthier, 1986; Sereno, 1991a; Langer, 2004). Within Saurischia, both branches of that clade, Sauropodomorpha and Theropoda, further elongated their necks (Galton & Upchurch, 2004). So sauropods inherited as their primitive state necks that were already more elongated, and heads that were proportionally smaller, than in most animals.

Within Theropoda, at least three lineages evolved especially long necks. The lengths of their necks can be estimated from their incomplete remains, though with some uncertainty, as follows.

Therizinosaurus cheloniformisMaleev, 1954 is a bizarre, long-necked giant theropod, known from incomplete remains. Measuring from Barsbold (1976, figure 1), its humerus was about 75 cm long. In a skeletal reconstruction of the therizinosauroid Nanshiungosaurus Dong, 1979 by Paul (1997, p. 145), the neck is 2.9 times the length of the humerus. If Therizinosaurus were similarly proportioned, its neck would have been about 2.2 m long.

Another giant theropod, Gigantoraptor erlianensis Xu et al., 2007 belongs to another long-necked group, Oviraptorosauria. Measured from the skeletal reconstruction of Xu et al. (2007, figure 1A), it appears to have had a neck 2.15 m in length – although this is conjectural as almost no cervical material is known.

Pterosaurs

Although it is often noted in general terms that azhdarchid pterosaurs had long necks (e.g., Howse, 1986; Witton & Naish, 2008), there are no published numeric estimates of neck length in this group. This is due to the lack of any published azhdarchid specimen with a complete neck (Witton & Habib, 2010): Quetzalcoatlus specimens at the Texas Memorial Museum may have complete necks, but have been embargoed since the early 1980s (Langston, 1981): a monographic description is still awaited. In the absence of a complete neck, all length estimates are uncertain, but it is nevertheless possible to arrive at an approximate length.

The azhdarchid for which the most complete neck has been described is Zhejiangopterus linhaiensis Cai & Wei, 1994, so we will base our estimates on this species. Cai & Wei (1994, table 7), give the lengths of cervicals 3–7 for three specimens, ZMNH M1323, M1324 and M1328. In all three, C5 is the longest cervical, as is generally true of pterodacyloid pterosaurs including azhdarchids (Howse, 1986, p. 323). Cai & Wei (1994) do not give lengths for C1 and C2, stating only that “the atlas-axis is completely fused and extremely short but morphological details are indistinct due to being obscured by the cranium” (p. 183, translation by Will Downs). Their figure 6, a reconstruction of Zhejiangopterus linhaiensis, bears this out, showing the atlas-axis as about one quarter the length of C3. Using this ratio to estimate the C1–2 lengths for each specimen, we find by adding the lengths of the individual cervicals that the three specimens had necks measuring approximately 511, 339 and 398 mm. These lengths are 3.60, 4.04 and 4.06 times the lengths of their respective C5s. On average, then, total C1–C7 neck length in known Zhejiangopterus specimens was about 3.85 times that of C5.

The azhdarchid Arambourgiania philadelphiae Arambourg, 1959 is the largest pterosaur for which cervical material has been described. Its type specimen, UJF VF1, is a single cervical vertebra. It was nearly complete when found, but has since been damaged and is now missing its central portion, but plaster replicas made before the damage indicate the extent of the missing portion. The preserved part of the vertebra was 620 mm long before the damage, and when complete it would have been about 780 mm long (Martill et al., 1998, p. 72). Assuming that the preserved element is C5, as considered likely by Howse (1986, p. 318) and Frey & Martill (1996, p. 240), the length of the C1–C7 region of the neck can be estimated as 3.85 times that length, which is 3.0 m.

Figure 3 Necks of long-necked sauropods, to scale.

Diplodocus, modified from elements in Hatcher (1901, plate 3), represents a “typical” long-necked sauropod, familiar from many mounted skeletons in museums. Puertasaurus, Sauroposeidon, Mamenchisaurus and Supersaurus modified from Scott Hartman’s reconstructions of Futalognkosaurus, Cedarosaurus, Mamenchisaurus and Supersaurus respectively. Alternating pink and blue bars are one meter in width. Inset shows Fig. 1 to the same scale.

Figure 4 Extent of soft tissue on ostrich and sauropod necks.

1, Ostrich neck in cross section from Wedel (2003, figure 2). Bone is white, air-spaces are black, and soft tissue is grey. 2, Hypothetical sauropod neck with similarly proportioned soft-tissue. (Diplodocus vertebra silhouette modified from Paul, 1997, figure 4A). The extent of soft tissue depicted greatly exceeds that shown in any published life restoration of a sauropod, and is unrealistic. 3, More realistic sauropod neck. It is not that the soft-tissue is reduced but that the vertebra within is enlarged, and mass is reduced by extensive pneumaticity in both the bone and the soft-tissue.

Figure 5 Simplified myology of the sauropod neck, in left lateral view, based primarily on homology with birds, modified from Wedel & Sanders (2002, figure 2).

Dashed arrows indicate muscle passing medially behind bone. A, B. Muscles inserting on the epipophyses, shown in red. C, D, E. Muscles inserting on the cervical ribs, shown in green. F, G. Muscles inserting on the neural spine, shown in blue. H. Muscles inserting on the ansa costotransversaria (“cervical rib loop”), shown in brown. Specifically: A. M. longus colli dorsalis. B. M. cervicalis ascendens. C. M. flexor colli lateralis. D. M. flexor colli medialis. E. M. longus colli ventralis. In birds, this muscle originates from the processes carotici, which are absent in the vertebrae of sauropods. F. Mm. intercristales. G. Mm. interspinales. H. Mm. intertransversarii. Vertebrae modified from Gilmore (1936, plate 24).

Figure 6 Basic cervical vertebral architecture in archosaurs.

1, Seventh cervical vertebra of a turkey, Meleagris gallopavo Linnaeus, 1758, traced from photographs by MPT. 2, Fifth cervical vertebra of the abelisaurid theropod Majungasaurus crenatissimus Depéret, 1896, UA 8678, traced from O’Connor (2007, figures 8 and 20). In these taxa, the epipophyses and cervical ribs are aligned with the expected vectors of muscular forces. The epipophyses are both larger and taller than the neural spine, as expected based on their mechanical importance. The posterior surface of the neurapophysis is covered by a large rugosity, which is interpreted as an interspinous ligament scar like that of birds (O’Connor, 2007). Because this scar covers the entire posterior surface of the neurapophysis, it leaves little room for muscle attachments to the spine. 3, Fifth cervical vertebra of Alligator mississippiensis Daudin, 1801, MCZ 81457, traced from 3D scans by Leon Claessens, courtesy of MCZ. Epipophyses are absent. 4, Eighth cervical vertebra of Giraffatitan brancai (Janensch, 1914) paralectotype MB.R.2181, traced from Janensch (1950, figures 43 and 46). Abbreviations: cr, cervical rib; e, epipophysis; ns, neural spine; poz, postzygapophysis.

The total number of cervical vertebrae in Zhejiangopterus is not clear: Cai & Wei (1994) imply that there are seven, and their illustrations (Figs. 5 and 6) indicate that in at least one specimen the vertebral column is complete. However, at least some azhdarchids seem to have have nine cervical vertebrae (e.g., Phosphatodraco, Pereda-Suberbiola et al., 2003), although the ninth “cervical” bears a long vertically oriented rib and must have contributed to the length of the torso rather than the neck. Bearing this in mind, the total neck length of Arambourgiania may have somewhat exceeded 3.0 m. In azhdarchids, C8 may be between 20% and 50% the length of C5 (Pereda-Suberbiola et al., 2003, p. 86), which might amount to 16–39 cm in Arambourgiania.

Another azhdarchid, Hatzegopteryx thambema Buffetaut, Grigorescu & Csiki (2002), may have been even larger than Arambourgiania, but no cervical material is known. Since its skull was much more robust that those of other azhdarchids (Buffetaut, Grigorescu & Csiki, 2002, p. 183), it was probably carried on a proportionally shorter and stronger neck.

Plesiosaurs

As marine reptiles, plesiosaurs benefited from the support of water and so lived under a wholly different biomechanical regime than terrestrial animals. The long necks of elasmosaurid plesiosaurs were constructed very differently from those of sauropods, consisting of many very short cervicals – 76 in the neck of Albertonectes vanderveldei Kubo, Mitchell & Henderson, 2012 and 71 in Elasmosaurus platyurus Cope, 1868 (Sachs, 2005, p. 92). Despite their marine lifestyle and very numerous cervicals, elasmosaurids did not attain neck lengths even half those of the longest-necked sauropods. According to Kubo, Mitchell & Henderson (2012, p. 570), “The approximately 7 m long neck of Albertonectes is the longest known for elasmosaurs (equal to 62% of total postcranial length).” Since the neck of Albertonectes was found articulated, the reported total neck length presumably includes the invertebral cartilage. Other elasmosaurs may have had equally long necks. The cervical vertebrae of Elasmosaurus platyurus holotype ANSP 10081 sum to 610.5 cm, based on individual vertebral lengths listed by Sachs (2005, p. 95). For other plesiosaurs, Evans (1993) estimated that the thickness of intercervical cartilage amounted to 14% of centrum length in Muraenosaurus Seeley, 1874 and 20% in Cryptoclidus Seeley, 1892. Using the average of 17% for Elasmosaurus, we can estimate its total neck length as 7.1 m (Fig. 2.4). This is within 6% of Leidy’s (1870) estimate of “almost twenty-two feet”, or about 6.7 m, and approximately equal to the 7-m neck length reported for Albertonectes by Kubo, Mitchell & Henderson (2012).

These longest necks are in the Diplodocus class. They lack most of the characters that we later identify as contributing to neck length in sauropods, but solved the support issue by being marine. We know from whales (see below) that a 7 m trachea need not impose the difficulties we might expect; but we don’t know whether sufficiently large lungs would fit inside an elasmosaur torso. So far, little work has been done on the physiological implications of neck elongation in plesiosaurs; more is needed.

Tanystropheus

The bizarre prolacertiform Tanystropheus merits mention as possessing perhaps the longest neck relative to torso length of any animal. The complete skeleton PIMUZ T 2818 has a total length of 420 cm (Nosotti, 2007, p. 8), of which the neck accounts for 211.2 cm (Tschanz, 1988, p. 1003) – almost exactly half. Nosotti (2007, p. 8) estimates the total length of the incomplete specimen PIMUZ T 2793 as 535 cm. If it were similarly proportioned to PIMUZ T 2818, its neck would have measured 269 cm.

Despite more than a century of study, there is no consensus on the habits or even environment of Tanystropheus. However, Nosotti (2007, p. 76) argues that limb proportions “unequivocally indicate the adaptation to an aquatic mode of life”. If this is correct, then its neck was subject to a quite different biomechanical regime than those of sauropods.

Sauropods

The necks of sauropod dinosaurs greatly exceeded in length those of all other animals (Wedel, 2006a). As noted above, they inherited long necks from their basal sauropodomorph ancestors. From this base, they elongated them yet further – both in ancestral forms and further still in more derived groups. Exceptionally long necks evolved in at least four distinct sauropod lineages (Fig. 3).

The basal eusauropod Mamenchisaurus Young, 1954 is known from several species. One, M. hochuanensis Young & Zhao, 1972, is known from an individual with a complete neck that is 9.5 m in length (personal measurement, MPT). Another species, M. sinocanadorum Russell & Zheng, 1993 is known only from skull elements and anterior cervical vertebrae, but by comparing this material with the corresponding elements of M. hochuanensis, its neck can be estimated to have been about 12 m long.

The known material of the diplodocid Supersaurus Jensen, 1985 includes a cervical vertebra whose centrum is 138 cm long. Comparing this with the lengths of similar vertebrae from the closely related Barosaurus Marsh, 1890, for which much more complete necks are known, suggests a complete neck length in the region of 15 m (Wedel, 2007a, p. 197).

The holotype and largest known specimen of Sauroposeidon Wedel, Cifelli & Sanders, 2000a consists of a sequence of four articulated cervical vertebrae, the largest of which has a centrum 125 cm long. The complete cervical series of the morphologically similar and possibly closely related brachiosaurid Giraffatitan Paul, 1988 is known, and consists of 13 cervicals measuring 8.5 m. The Sauroposeidon cervicals are on average 37% longer than the corresponding vertebrae of Giraffatitan, suggesting a complete neck length of about 11.5 m. If Sauroposeidon is a basal somphospondyl rather than a brachiosaurid, as suggested by D’Emic & Foreman (2012), then a more apposite comparison might be to Euhelopus, which had 17 cervicals. The complete cervical series of Euhelopus PMU R233 is 13.2 times the length of the longest cervical (3765 mm vs 285 mm; Wiman, 1929). Applying a similar scaling relationship to Sauroposeidon, and conservatively assuming that the largest available vertebra was the longest in the neck, yields an estimated neck length of 16.5 m. We will not know which of the two estimates is more accurate until more articulated cervical material of Sauroposeidon comes to light.

PuertasaurusNovas et al., 2005 is the largest titanosaur for which cervical material has been described. The single known cervical vertebra is 118 cm in total length, including overhanging prezygapophyses, and its incomplete centrum can be reconstructed after related titanosaurs as having been 105 cm long. Cross-scaling with Malawisaurus Jacobs et al., 1993, which has the most similar cervical vertebrae among titanosaurs known from complete cervical material, yields a total neck length of 9 m.

Table 1 lists a selection of sauropods, mostly known from complete or nearly complete necks, showing how they vary in length, cervical count, centrum length, cervical rib length, and maximum elongation index.

Factors Enabling Long Necks

Discounting the aquatic plesiosaurs, whose necks were subject to different forces from those of terrestrial animals, neck-length limits in the range of two to three meters seem to apply to every group except sauropods, which exceeded this limit by a factor of five. Whatever mechanical barriers prevented the evolution of truly long necks in other terrestrial vertebrates, sauropods did not just break that barrier – they smashed it. Since four separate sauropod lineages evolved necks three or four times longer than those of any of their rivals, it seems likely that sauropods shared a suite of features that facilitated the evolution of such long necks. What were these features?

Absolutely large body size

It is obviously impossible for a terrestrial animal with a torso the size of a giraffe’s to carry a 10 m neck. Sheer size is probably a necessary, but not sufficient, condition for evolving an absolutely long neck. Mere isometric scaling would of course suffice for larger animals to have longer necks, but Parrish (2006, p. 213) found a stronger result: that neck length is positively allometric with respect to body size in sauropods, varying with torso length to the power 1.35. This suggests that the necks of super-giant sauropods may have been even longer than imagined: Carpenter (2006, p. 133) estimated the neck length of the apocryphal giant Amphicoelias fragillimus Cope, 1878 as 16.75 m, 2.21 times the length of 7.5 m used for Diplodocus, but if Parrish’s allometric curve pertained then the true value would have been 2.211.35 = 2.92 times as long as the neck of Diplodocus, or 21.9 m; and the longest single vertebra would have been 187 cm long.

The allometric equation of Parrish (2006) is descriptive, but does not in itself suggest a causal link between size and neck length. As noted by Wedel, Cifelli & Sanders (2000b, p. 377), one possible explanation is that, because of their size, sauropods were under strong selection for larger feeding envelopes, which drove them to evolve longer necks.

Quadrupedal stance

One of the key innovations in the evolution of sauropods was quadrupedality, facilitated by characters such as forelimb elongation, columnar limbs and short metapodials (Wilson & Sereno, 1998, p. 24). As well as providing a platform for the evolution of large body size, the stability of the quadrupedal posture also enabled the evolution of longer necks: although progressive elongation displaced the centre of mass forwards from above the hindlimbs, it remained in the stable region between fore and hindlimbs.

Computer modelling shows that theropod dinosaurs such as Tyrannosaurus rex Osborn, 1905 attained masses of 7 or even 10 tonnes (Hutchinson et al., 2011), and other giant theropods including Therizinosaurus and Gigantoraptor were probably of comparable size. However, they did not evolve necks as long as those of sauropods with similar mass, probably in part for this reason: the increased moment caused by neck elongation in a biped must be counteracted by an equal moment caused by a longer or more massive tail, increasing the physiological cost.

Small head

Sauropods inherited proportionally small heads from ancestral sauropodomorphs, and continued to reduce their proportional size. In many clades, they were further lightened by reduced dentition, because unlike other large-bodied animals such as hadrosaurs, ceratopsians and elephants, sauropods did not orally process their food. Sauropod heads were simple cropping devices with a brain and sense organs, and did not require special equipment for obtaining food, such as the long beaks of azhdarchids (Chure et al., 2010, pp. 388–389). The reduction in head weight would have reduced the required lifting power of the necks that carried them, and therefore the muscle and ligament mass could be reduced, allowing the necks to be longer than would have been possible with heavier heads. Other groups of large-bodied animals have not evolved long necks, instead either developing large heads on short necks (ceratopsians, proboscideans, tyrannosaurs) or a compromise of a medium-sized head on a medium-length neck (hadrosaurs, indricotheres). It is significant that all other clades of large (>10 ton) terrestrial herbivores – ceratopsians, hadrosaurs, proboscideans, and indricotheres – practiced extensive oral processing of their food, requiring massive dentition and correspondingly large heads.

Numerous cervical vertebrae

Many groups of animals seem to be constrained as to the number of cervical vertebrae they can evolve. With the exceptions of sloths and sirenians, mammals are all limited to exactly seven cervicals due to developmental constraints: mutations to the Hox genes that control the number of cervicals also give rise to neonatal cancer and other birth defects (Galis, 1999; Galis & Metz, 2003). Azdarchids are variously reported as having seven to nine cervical vertebrae, but never more; non-avian theropods do not seem to have exceeded the 13 or perhaps 14 cervicals of Neimongosaurus Zhang et al., 2001, with eleven or fewer being more typical.

By contrast, sauropods repeatedly increased the number of their cervical vertebrae, attaining as many as 19 in Mamenchisaurus hochuanensis (Young & Zhao, 1972, p. 3–7). Modern swans have up to 25 cervical vertebrae, and as noted above the marine reptile Albertonectes had 76 cervical vertebrae. Multiplication of cervical vertebrae obviously contributes to neck elongation.

Elongate cervical vertebrae

The shape of cervical vertebrae is quantified by the elongation index (EI), defined by Wedel, Cifelli & Sanders (2000b, p. 346) as the anteroposterior length of the centrum divided by the midline height of its posterior articular face. As shown in Table 1, EI in sauropods routinely exceeded 4.0, and in some cases exceeded 6.0: Sauroposeidon C6 attained 6.1, and Erketu Ksepka & Norell, 2006 C5 attained 7.0.

A similar degree of elongation is approached by the ostrich, in which C12 can attain an EI of 4.4 (measured from Mivart, 1874, figure 29), and by the giraffe, in which the axis can attain an EI of 4.71 (personal measurement of FMNH 34426). It is greatly exceeded by azhdarchid pterosaurs, among which C5 of Quetzalcoatlus Lawson, 1975 can attain an astonishing 12.4 (measured from Witton & Naish (2008, figure 4c)) and an isolated cervical from the Hell Creek Formation might have achieved 15 (measured from Henderson & Peterson (2006, figure 3)).

But other long-necked groups are more limited in their elongation of individual vertebrae. Paraceratherium seems have been limited to about 3.3 for C2 (measured from Granger & Gregory (1936, figure 7)) and much less for the other vertebrae. Elongation indexes of therizinosaurs such as Therizinosaurus probably did not greatly exceed 1.0 (measured for Nanshiungosaurus from Dong (1979, figures 1–2)); those of oviraptorosaurs such as Gigantoraptor, 2.0 (measured from a photograph by MJW of referred specimen IGM 100/1002 of Khaan mckennai Clark, Norell & Barsbold, 2001). The very numerous vertebrae of Elasmosaurus are not very elongate, mostly having an EI around 1.0 and not exceeding about 1.4 (measured from Sachs (2005), figure 4).

Figure 7 Disparity of sauropod cervical vertebrae.

1, Apatosaurus “laticollis” Marsh, 1879b holotype YPM 1861, cervical ?13, now referred to Apatosaurus ajax (see McIntosh, 1995), in posterior and left lateral views, after Ostrom & McIntosh (1966, plate 15); the portion reconstructed in plaster (Barbour, 1890, figure 1) is grayed out in posterior view; lateral view reconstructed after Apatosaurus louisae (Gilmore, 1936, plate XXIV). 2, “Brontosaurus excelsus” Marsh, 1879a holotype YPM 1980, cervical 8, now referred to Apatosaurus excelsus (see Riggs, 1903), in anterior and left lateral views, after Ostrom & McIntosh (1966, plate 12); lateral view reconstructed after Apatosaurus louisae (Gilmore, 1936, plate XXIV). 3, “Titanosaurus” colberti Jain & Bandyopadhyay, 1997 holotype ISIR 335/2, mid-cervical vertebra, now referred to Isisaurus (See Wilson & Upchurch, 2003), in posterior and left lateral views, after Jain & Bandyopadhyay (1997, figure 4). 4, “Brachiosaurus” brancai paralectotype MB.R.2181, cervical 8, now referred to Giraffatitan (see Taylor, 2009), in posterior and left lateral views, modified from Janensch (1950, figures 43–46). 5, Erketu ellisoni holotype IGM 100/1803, cervical 4 in anterior and left lateral views, modified from Ksepka & Norell (2006, figures 5a–d).

Air-sac system

One limiting factor on neck length is the difficulty of breathing through a long trachea. If the trachea is narrow, then it is difficult to inhale sufficient air quickly – a problem exacerbated by friction of inhaled air against the tracheal wall. But if the trachea is wider, its volume is increased, and a larger quantity of used air in the “tracheal dead space” is re-inhaled in each breath, reducing the oxygen content of each breath.

For this reason, it would be reasonable to expect animals to evolve the shortest possible trachea. However, in one clade – birds – an elongate trachea is not unusual, having evolved in swans (Banko, 1960), cranes (Johnsgard, 1983), moas (Worthy & Holdaway, 2002), birds-of-paradise (Frith, 1994) and several other groups. This trend reaches its peak in the trumpet manucode Phonygammus keraudrrenii (Clench, 1978). In some mature males, the trachea coils back on itself so many times that its total length exceeds 800 mm, nearly three times the total body length of approx. 30 cm. Alone among extant animals, birds are able to cope with such extreme tracheal elongation, due to their very efficient lungs and the large tidal volume of the whole respiratory system on account of the voluminous air-sacs.

It is now well established that sauropods had an air-sac system similar to that of extant birds (Wedel, 2003), and most likely a similarly efficient flow-through lung (Wedel, 2009). These features would have greatly eased the problem of tracheal dead space, facilitating the evolution of longer necks. The air-sac system, including cervical air-sacs and extensive cervical diverticula running the full length of the neck, would also have served to lighten long necks.

Among other long-necked animals, theropods (including Therizinosaurus and Gigantoraptor) and pterosaurs also had air-sac systems; but the mammals (giraffes, Paraceratherium) did not. However, whales provide an example suggesting it is unlikely that the evolution of long necks in terrestrial mammals has been limited by tracheal dead space. In a male sperm whale (Physeter) with a total body length of 16 m, the length of the head is 5.6 m (Nishiwaki, Ohsumi & Maeda (1963), cited in Cranford (1999, table 1)). The largest sperm whales are up to 20 m in total body length (Gosho, Rice & Breiwick, 1984), which would give a head length of 7 m if these largest individuals scaled isometrically with the 16-m whales. However, the head length of sperm whales is positively allometric and increases with age even in adults (Cranford, 1999, p. 1141 and figure 4), so a 20-m adult might well have a head slightly more than 7 m long. As in all cetaceans, the skull of a sperm whale is separated from the ribcage by the highly compressed cervical series. Finally, the nasal airways in sperm whales do not take a direct path from the blowhole to the lungs but describe sinuous curves through the head (Cranford, 1999, figures 1 and 3). In a sperm whale with a 7-m head, the internal convolution of the nasal airways and the addition of the trachea spanning from the head to the trunk would give the path from blowhole to lungs a total length of perhaps 9 m, showing that tracheae at least that long are possible without an air sac system.

Vertebral architecture

Aside from the factors previously discussed, the elongation of sauropod necks was made possible by the distinctive architecture of their cervical vertebrae. The various aspects of their architecture are discussed in detail in the next section.

Architecture of Sauropod Necks

Pneumaticity of cervical vertebrae

Not only did sauropods have a soft-tissue diverticular system, but the diverticula often invaded the vertebrae, leaving extensive excavations and other traces (e.g., Janensch, 1947; Wedel, Cifelli & Sanders, 2000b). Indeed, it is from the latter that we are able to infer the former.

The air space proportion (ASP) of a bone is the proportion of its volume taken up by pneumatic cavities (Wedel, 2005). Dicraeosaurids (Dicraeosaurus, Amargasaurus, and related taxa) had reduced postcranial pneumaticity compared to other neosauropods, both in terms of the number of presacral vertebrae that were pneumatized, and in the air space proportion (Schwarz & Fritsch, 2006). The presacral vertebrae of most neosauropod taxa had ASPs between 0.50 and 0.70 (Table 2) – as lightly built as the pneumatic bones of most birds (Wedel, 2005). Basal sauropods outside or near the base of Neosauropoda, such as Cetiosaurus, Jobaria, and Haplocanthosaurus, had much lower ASPs, around 0.40. (ASPs of Cetiosaurus and Jobaria are estimates based on personal observations of the holotypes and referred specimens, and comparisons to CT scans of similarly-constructed Haplocanthosaurus vertebrae.)

Table 2 Air Space Proportion (ASP) of sections through sauropod vertebrae.

Measurements are taken from CT sections, photographs, and published images. Sections are transverse unless otherwise noted. Although this dataset is almost three times as large as that reported by Wedel (2005), the mean is the about same, 0.61 compared to 0.60. Abbreviations: C, cervical; Cd, caudal; D, dorsal; P, presacral.

Taxon	Region		ASP	Source	
Apatosaurus	C	condyle	0.69	OMNH 01094	
		mid-centrum	0.52	,,	
		posterior centrum	0.73	,,	
		cotyle	0.32	,,	
	C	condyle	0.63	OMNH 01340	
		mid-centrum	0.69	,,	
		cotyle	0.49	,,	
	C	condyle	0.52	CM 555 C6	
		mid-centrum	0.75	,,	
		posterior centrum	0.59	,,	
		cotyle	0.34	,,	
	C	parapophysis	0.6	BYU 11998	
	C	cotyle	0.7	BYU 11889	
Brachiosaurus	C	condyle	0.55	BYU 12866	
		mid-centrum	0.67	,,	
		posterior centrum	0.81	,,	
Brachiosauridae	C	mid-centrum	0.89	MIWG 7306	
	P		0.65	Naish & Martill (2001, plate 32)	
	P		0.85	Naish & Martill (2001, plate 33)	
	P		0.85	MIWG uncatalogued	
Camarasaurus	C	condyle	0.51	OMNH 01109	
		mid-centrum	0.68	,,	
		cotyle	0.54	,,	
	C	condyle	0.49	OMNH 01313	
		mid-centrum	0.52	,,	
		cotyle	0.5	,,	
	D	mid-centrum	0.58	Ostrom & McIntosh (1966, plate 23)	
	D	mid-centrum	0.63	Ostrom & McIntosh (1966, plate 23)	
	D	mid-centrum	0.71	Ostrom & McIntosh (1966, plate 23)	
Chondrosteosaurus	P	centrum (horiz.)	0.7	Naish & Martill (2001, figure 8.5)	
Diplodocus	C	condyle	0.56	BYU 12613	
		mid-centrum	0.54	,,	
		posterior centrum	0.66	,,	
Giraffatitan	C	condyle	0.73	Janensch (1950, figure 70)	
	C	condyle (sagittal)	0.57	Janensch (1947, figure 4)	
	D	mid-centrum	0.59	Janensch (1947, figure 2)	
Haplocanthosaurus	C	condyle	0.39	CM 879-7	
		mid-centrum	0.56	,,	
		posterior centrum	0.42	,,	
		cotyle	0.28	,,	
	D	mid-centrum	0.36	CM 572	
Malawisaurus	C	condyle	0.56	MAL-280-1	
		mid-centrum	0.62	,,	
	C	condyle	0.57	MAL-280-4	
		mid-centrum	0.56	,,	
Phuwiangosaurus	C	mid-centrum	0.55	Martin (1994, figure 2)	
Pleurocoelus	C	mid-centrum	0.55	Lull (1911, plate 15)	
Saltasaurus	D	centrum (horiz.)	0.62	Powell (1992, figure 16)	
		mid-centrum	0.55	,,	
		neural spine (horiz.)	0.82	,,	
	D	prezygapophysis	0.78	Powell (1992, figure 16)	
Sauroposeidon	C	prezyg. ramus	0.89	OMNH 53062	
		postzygapophysis	0.74	,,	
		anterior centrum	0.75	,,	
Supersaurus	C	mid-centrum	0.64	WDC-DMJ021	
Tornieria	C	mid-centrum	0.56	Janensch (1947, figure 8)	
	C	posterior centrum	0.77	Janensch (1947, figure 3)	
	D	condyle (sagittal)	0.78	Janensch (1947, figure 9)	
	Cd	mid-centrum	0.47	Janensch (1947, figure 7)	
Sauropoda indet.	C	mid-centrum	0.54	OMNH 01866	
	C	posterior centrum	0.46	OMNH 01867	
	C	mid-centrum	0.55	OMNH 01882	
					
	MEAN		0.61		

The effects of pneumatization on the mass of the cervical series have been little explored. The centrum walls, laminae, septae, and struts that comprised the vertebrae were primarily made of compact bone (Reid, 1996). The specific gravity (SG) of compact bone is 1.8–2.0 in most tetrapods (Spector, 1956), so an element with an ASP of 0.60 (and therefore a compact bone proportion of 0.40) would have an in-vivo SG of 0.7–0.8. Some sauropod vertebrae were much lighter. For example, Sauroposeidon has ASP values up to 0.89 and therefore SG as low as 0.2 in some parts of its vertebrae. On the other hand, many basal sauropods had ASPs of 0.30–0.40 and therefore SG of 1.1–1.4.

An important effect of postcranial pneumaticity is to broaden the range of available densities in skeletal construction. Animals without postcranial pneumaticity, including mammals and ornithischian dinosaurs, are constrained to build their skeletons out of bone tissue (SG = 1.8–2.0) and marrow (SG = 0.93; Currey & Alexander, 1985, p. 455). Therefore, the whole-element densities of their postcranial bones will always be between 1.0 and 2.0; they cannot be more dense than bone tissue, nor can they be constructed entirely out of marrow. The pneumatic bones of pterosaurs and saurischian dinosaurs are made of bone tissue (SG = 1.8–2.0) and air space (SG = 0), which allows them to have whole-element densities that are much lower. The lightest postcranial bones for which data are available are those of Sauroposeidon and some pterosaurs. The cranial bones of some birds are even lighter. Seki, Schneider & Meyers (2005) reported an SG of 0.05 for the “bone foam” inside the beak of the toucan (Rhamphastos toco), and an SG of 0.1 for the entire beak. To date, this is the lightest form of bone known in any vertebrate.

While the impact of soft-tissue diverticula is more difficult to assess, it is easy to imagine that the density of a typical neosauropod neck may have been less than 0.5 kg/dm3. Although pneumaticity was undoubtedly an important adaptation for increasing the length of the neck without greatly increasing its mass, a longer neck remains more mechanically demanding than a shorter neck of the same mass, because that mass acts further from the fulcrum of the cervicodorsal joint, increasing the moment that must be counteracted by the epaxial tension members. Also, longer trachea and blood vessels cause physiological difficulties: weight support is only one of the problems imposed by a long neck.

While pneumaticity may be necessary for the development of a long neck, it is clearly not sufficient: while three groups of theropods, all pneumatic, evolved necks in the 2–2.5 m range, and pneumatic pterosaurs attained 3 m, these remain well short of even the less impressive sauropod necks (e.g., 4 m in Camarasaurus AMNH 5761; Osborn & Mook, 1921).

Extent of soft-tissue relative to size of vertebrae

In most extant vertebrates including birds and crocodilians, the diameter of the neck is three or four times that of the cervical vertebrae that form its core. Even in long, thin-necked animals such as the ostrich, the muscular part of the neck is twice as wide and 2.3 times as tall as the enclosed vertebra (Fig. 4.1), and if the trachea and skin and related soft-tissue is included the dorsoventral thickness of the neck is fully 3.3 times that of the vertebra alone (Dzemski & Christian, 2007, figure 2). (In the caption to Wedel (2003, figure 2), from which Fig. 4.1 of the present paper is modified, the small airspace ventral to the vertebra was misidentified as the trachea. In fact it is a complex of diverticula around the carotid arteries.)

If the necks of sauropods were as heavily muscled as those of ostriches, then they would have appeared in cross section as shown in Fig. 4.2. But life restorations of sauropods going back to the 1800s have been unanimous that this cannot have been the case in sauropods, as such over-muscled necks would have been too heavy to lift; and the various published reconstructions of sauropod neck cross sections (e.g., Paul, 1997, figure 4; Schwarz, Frey & Meyer, 2007, figure 7, 8A, 9E) all agree in making the total diameter including soft-tissue only 105–125% that of the vertebrae alone.

This is a consequence of scaling, which makes it impossible for sauropod necks to be similar to those of ostriches. Consider an ostrich neck scaled up by a linear factor of L. The weight exerted by the neck is proportional to L3 but the cross-sectional area of the bracing members is proportional to only L2. Stress is force/area, which is proportional to L3/L2 = L, so the stress on the bracing members that support the neck varies linearly with L. (The weight of the neck acts at a distance proportional to L from the torso, and the bracing members acts at a distance proportional to L above the neck-torso articulation, so these factors cancel out of the balancing moment equation.) Since isometric similarity is precluded the necks of sauropods had to be re-engineered in order to attain such great sizes. Can that have been done by reducing the amount of muscle?

In fact, comparing the restored neck of a sauropod with that of an ostrich scaled to the same body size, it is apparent that the sauropod neck has not so much reduced the size of the neck muscles as increased the size of the cervicals vertebrae themselves (Fig. 4.3): they are much larger compared to the torso than in the ostrich. Simply increasing the size of the vertebrae would not be a good strategy for neck support, because bone is the densest material in the body apart from tooth enamel and dentine. But as noted above, sauropod vertebrae were very pneumatic, typically consisting of 60% air. In effect, sauropods inflated their vertebrae within the muscular envelope of the neck, moving the bone, muscle and ligament away from the centre so that they acted with greater mechanical advantage: higher epaxial tension members, lower hypaxial compression members, and more laterally positioned paraxials.

Muscle attachments

In extant animals, the mechanically significant soft tissues of the neck (muscles, tendons and ligaments) can be examined and their osteological correlates identified. In extinct animals, except in a very few cases of exceptional preservation, only the fossilized bones are available: but using extant animals as guides, osteological features can be interpreted as correlates of the absent soft tissue, so that the ligaments and musculature of the extinct animal can be tentatively reconstructed (Bryant & Russell, 1992; Witmer, 1995). In order to do this for sauropods, it is necessary first to examine their extant outgroups, the birds and crocodilians.

In all vertebrates, axial musculature is divided both into left and right sides and into epaxial and hypaxial (i.e., dorsal and ventral to the vertebral column) domains, yielding four quadrants. In birds, the largest and mechanically most important epaxial muscles (M. longus colli dorsalis and M. cervicalis ascendens) insert on the epipophyses of the cervical vertebrae – that is, distinct dorsally projecting tubercles above the postzygapophyses. The large hypaxial muscles (M. flexor colli lateralis, M. flexor colli medialis, and M. longus colli ventralis) insert on the cervical ribs (Fig. 5; Baumel et al., 1993; Tsuihiji, 2004). The osteology of the cervical vertebrae makes mechanical sense; the major muscle insertions are prominent osteological features located at the four “corners” of the vertebrae (Fig. 6.1). Non-avian theropods resembled birds in this respect, having prominent epipophyses and sizable cervical ribs, which point in the four expected directions (Fig. 6.2).

The cervical architecture is rather different in crocodilians, and in non-archosaurian diapsids such as lizards, snakes, ichthyosaurs and plesiosaurs: there are no epipophyses, and the main epaxial neck muscles are the Mm. Interspinales, which attach to the neural spines rather than to epipophyses (Fig. 6.3). In most sauropods, the cervical vertebrae do have epipophyses, but the neural spines are as prominent or more so (Fig. 6.4). In this respect, sauropod osteology is intermediate between the conditions of crocodilians and birds – so the widely recognized similarity of sauropod cervicals to those of birds (e.g., Wedel & Sanders, 2002; Tsuihiji, 2004), while significant, should not be accepted unreservedly. Since the prominent neural spine serves as the primary attachment site for epaxial muscles in most theropod outgroups, the condition in birds and other theropods is derived; that of sauropods retains aspects of the primitive condition.

Although sauropods shared a common bauplan, their morphological disparity was much greater than has usually been assumed (Taylor & Naish, 2007, pp. 1560–1561). This disparity is particularly evident in the cervical vertebrae (Fig. 7). Those of Apatosaurus Marsh, 1877, for example, are anteroposteriorly short and dorsoventrally tall, and have short, robust cervical ribs mounted far ventral to the centra; the cervical centra of Isisaurus colberti (Jain & Bandyopadhyay, 1997) are even shorter anteroposteriorly, but have more dorsally located cervical ribs; by contrast, the cervical vertebrae of Erketu ellisoni Ksepka & Norell, 2006 are relatively much longer and lower, and have long, thin cervical ribs mounted only slightly ventral to the centra, which are sigmoid rather than cylindrical. Towards the middle ground of these extremes fall the cervical vertebrae of Giraffatitan, which are anteroposteriorly longer and dorsoventrally shorter than those of Apatosaurus, but not as anteroposteriorly long or as dorsoventrally short as those of Erketu. In light of the demanding mechanical constraints that were imposed on sauropods, it is surprising that their necks vary so much morphologically, with different lineages having evolved dramatically different solutions to the problem of neck elongation and elevation.

Interpretation of sauropods as living animals is made especially difficult by the lack of good extant analogues. Among animals with long necks, giraffes, camels, and other artiodactyls have very different cervical osteology and (we assume) myology; and even the longest of their necks, at about 2.4 m, is only one sixth the length attained by some sauropods. Birds are phylogenetically closer to sauropods, and some birds (e.g., swans and ostriches) have proportionally very long necks. Furthermore, the presence in most sauropods of epipophyses similar to those of birds suggests that sauropods were myologically similar to birds. However, the small absolute size of birds means that the forces acting on their necks are so different that we can’t assume that sauropod necks functioned in the same ways – just as the problems involved in flight through air for high-Reynolds number fliers such as birds are very different than than they are for low-Reynolds number fliers such as fruit-flies, whose aerodynamics are dominated by friction drag rather than form drag.

Because sauropods were so much bigger than their relatives, and their necks so much longer, mechanical considerations in the construction of their necks were significantly more important than in their outgroups. Furthermore, the great size and shape disparity between sauropods and their outgroups means that interpretations of cervical soft-tissue anatomy in sauropods cannot be based purely on the extant phylogenetic bracket method: this alone would be no more informative than trying to determine the anatomy of elephants from that of manatees and hyraxes.

With all these caveats in mind, the best extant analogues for sauropod necks nevertheless remain those of birds: they are the only extant animals that share with sauropods epipophyses above their postzygapophyses, pronounced cervical ribs, and pneumatic foramina (Figs. 6.1 and 6.4). The first two of these features were inherited from a common saurischian ancestor. The foramina seem to have been independently derived in birds, but this was possible because air sacs and soft-tissue pneumatic diverticula were likely present in the common saurischian ancestor (Wedel, 2006b; Wedel, 2007b). These observations enable us to draw conclusions about sauropod neck soft tissue beyond what the extant phylogenetic bracket would allow. Specifically, the epipophyses are osteological correlates of the M. longus colli dorsalis and M. cervicalis ascendens epaxial muscles, which must therefore have been present in sauropods, although we can not conclude from this that they were necessarily the dominant epaxial muscles as they are in birds.

Neural spines

The neural spines and epipophyses of sauropods both anchored epaxial muscles, but as they were differently developed in different taxa, they were probably of varying mechanical importance in different taxa. For example, based on their relative heights, epipophyses probably dominated neural spines in Apatosaurus (Fig. 7.1) but neural spines may have dominated in Isisaurus Wilson & Upchurch, 2003 and Giraffatitan (Figs. 7.3 and 7.4). In some sauropods, including Erketu and Mamenchisaurus, which were proportionally long-necked even by sauropod standards, the neural spines are strikingly low, and the epipophyses no higher – a surprising arrangement, as low spines would have reduced the lever arm with which the epaxial tension members worked. Among these sauropods with low neural spines, some have rugose neurapophyses with spurs directed anteriorly and posteriorly from the tip of the spine (Fig. 8). These appear either to have anchored discontinuous interspinous ligaments, as found in all birds (see Wedel, Cifelli & Sanders, 2000b, figure 20), or to have been embedded in a continuous supraspinous ligament, as found in the ostrich (Dzemski & Christian, 2007, pp. 701–702).

Figure 8 Sauropod cervical vertebrae showing anteriorly and posteriorly directed spurs projecting from neurapophyses.

1, Cervical 5 of Sauroposeidon holotype OMNH 53062 in right lateral view, photograph by MJW. 2, Cervical 9 of Mamenchisaurus hochuanensis holotype CCG V 20401 in left lateral view, reversed, from photograph by MPT. 3, Cervical 7 or 8 of Omeisaurus junghsiensis Young, 1939 holotype in right lateral view, after Young (1939, figure 2). (No specimen number was assigned to this material, which has since been lost. DWE Hone personal communication 2008.)

In some sauropods, the cervical neural spines are bifid (i.e., having separate left and right metapophyses and a trough between them). This morphology appears to have evolved at least five times (in Mamenchisaurus, flagellicaudatans, Camarasaurus Cope, 1877, Euhelopodidae sensu D’Emic (2012) and Opisthocoelicaudia Borsuk-Bialynicka, 1977) with no apparent reversals. This morphology, then, seems to have been easy for sauropods to gain, but difficult or perhaps impossible to lose. Bifid cervical vertebrae are extremely uncommon in other taxa, and among extant animals they are found only in birds: the ibis Theristicus (Tambussi et al., 2012) and ratites including Rhea americana Linnaeus, 1758 (Tsuihiji, 2004, figure 2B), Casuarius casuarius Brisson, 1760 (Schwarz, Frey & Meyer, 2007, figure 5B) and Dromaius novaehollandiae Latham, 1790 (Osborn, 1898, figure 1). It has often been assumed that in sauropods with bifid cervical spines, the intermetapophyseal trough housed a large ligament analogous to the nuchal ligament of artiodactyl mammals (e.g., Janensch, 1929, plate 4; Alexander, 1985, pp. 13–14; Wilson & Sereno, 1998, p. 60). Such an arrangement seems unlikely, as lowering the ligament into the trough would reduce its mechanical advantage; however, this is similar to the arrangement seen in Rhea americana, in which branches of the “nuchal ligament” attach to the base of the trough (Tsuihiji, 2004, figure 3). More direct evidence is found in ligament scars in the troughs of some diplodocids: these can be prominent, as in the doorknob-sized attachment site in the Apatosaurus sp. cervical OMNH 01341 (Fig. 9.1).

Figure 9 Ligament scars and pneumatic foramina in intermetapophyseal troughs.

Bifid presacral vertebrae of sauropods showing ligament scars and pneumatic foramina in the intermetapophyseal trough. 1, Apatosaurus sp. cervical vertebra OMNH 01341 in right posterodorsolateral view, photograph by MJW. 2, Camarasaurus sp. dorsal vertebrae CM 584 in dorsal view, photograph by MJW. Abbreviations: las, ligament attachment site; pfa, pneumatic fossa; pfo, pneumatic foramen.

However, ligament cannot have filled the trough as envisaged by Alexander (1985, figure 4C), because pneumatic foramina are often found in the base of the troughs of presacral vertebrae, for example in the cervicals of Apatosaurus (Fig. 9.1) and the dorsal vertebrae of Camarasaurus sp. CM 584 (Fig. 9.2). In some specimens, a ligament scar and pneumatic foramen occur together in the intermetapophyseal trough (Fig. 9.1; Schwarz, Frey & Meyer, 2007, figure 6E). Pneumatic diverticula are sometimes found between the centropostzygapophyseal laminae even in sauropods with non-bifid spines, as shown by the isolated brachiosaurid cervical MIWG 7306 from the Isle of Wight (Naish, 2008), so the presence of soft-tissue diverticula in this location is probably primitive for Neosauropoda at least. In conclusion, while some ligament was undoubtedly present within the trough formed by the metapophyses of bifid neural spines, much of the space was probably filled with pneumatic diverticula.

One possible advantage of bifid spines would be to increase the lateral leverage of the ligaments and muscles that attach to the metapophyses, enabling them to contribute to lateral stabilisation and motion as well as vertical. A cantilevered beam, which is what a sauropod neck is in mechanical terms, requires only a single dorsal tension member to stabilize it vertically, but two (one on each side) to stabilize it horizontally. A sauropod neck that was supported from above only by a single midline tension member would need additional horizontal stabilization from muscles and ligaments not directly involved in support.

Whatever the advantages of bifid spines, they were clearly not indispensable, as some sauropod lineages evolved very long necks with unsplit spines (e.g., brachiosaurids, Sauroposeidon, and most titanosaurs, including the very long-necked Puertasaurus). Even in taxa that do have bifid spines, they are rarely split through the whole series: for example, the first eight cervicals of Barosaurus do not have bifid spines (McIntosh, 2005; MJW, pers. obs). Even in Camarasaurus lewisi BYU 9047, in which every postaxial cervical vertebra is at least partially bifid (McIntosh et al., 1996b), the bifurcation is very slight in the anterior cervicals and probably of little mechanical consequence. If bifid spines conferred a great advantage, they would presumably be found throughout the neck – although the importance of stability, and the difficulty of attaining it, is greater in the posterior part of the neck, which bears greater forces than the anterior part. Since bifid spines always occur together with unsplit spines, it seems likely that however they were used mechanically, it was probably not radically different from neural spine function in vertebrae with unsplit spines.

Epipophyses

As noted above, the epipophyses are the insertion points of the largest and longest epaxial muscles in birds, whereas in crocodilians the epipophyses are non-existent, and no major muscles insert above the postzygapophyses (Tsuihiji, 2004). Epipophyses are found in most, though not all, sauropods and theropods. For example, they are absent in the titanosaurs Malawisaurus (pers. obs., MJW; Gomani, 2005, figure 8) and Isisaurus (Fig. 7.3); but their presence in other titanosaurs such as Rapetosaurus Curry & Forster, 2001  (Curry & Forster, 2001, figure 3A), and Saltasaurus Bonaparte & Powell, 1980 (Powell, 1992, figure 5) and in outgroups such as Giraffatitan (Fig. 7.4) and Camarasaurus (Osborn & Mook, 1921, plate LXVII, figure 9; McIntosh et al., 1996a, figure 29) indicates that their absences in Malawisaurus and Isisaurus, if not due to damage to the material, represent secondary losses. Not all muscles leave diagnostic traces on the skeleton, so the absence of epipophyses does not mean that the epaxial muscles that insert above the postzygapophyses were absent. It is worth noting that the available material of Malawisaurus and Isisaurus pertains to relatively small individuals; perhaps the forces exerted by the epaxial muscles were not enough to produce distinctive scarring of the bone that we would recognize as epipophyses.

The existence of epipophyses on the cervical vertebrae of most sauropods, together with those in theropods and birds, suggests that epaxial muscles were inserting above the postzygapophyses at least by the origin of Saurischia. Epipophyses are also known in basal ornithischians, e.g., Lesothosaurus Galton, 1978 (Sereno, 1991b, figure 8A) and Heterodontosaurus Crompton & Charig, 1962 (Santa Luca, 1980, figure 5A), and also in the basal pterosaur Rhamphorhynchus Meyer, 1846 (Bonde & Christiansen, 2003, figures 6–9), suggesting that these insertion points were in use at the base of Dinosauria and possibly Ornithodira.

In sauropods, the size and location of the epipophyses is variable: in C8 of Giraffatitan, the epipophyses are approximately half as high above the centrum as the neurapophysis (Fig. 7.4); in anterior cervicals of Erketu, the epipophyses are equally as high as the tips of the neural spines (Fig. 7.5), although the spines are higher in posterior cervicals. It is possible that in the posterior cervicals of some Apatosaurus ajax Marsh, 1877 specimens, the epipophyses are higher than the metapophyses (Fig. 7.1), but it is difficult to be sure as the vertebrae that seem to most closely approach this condition are at least partly reconstructed in plaster (Barbour, 1890, figure 1). In any event, it is clear from preserved sequences of Apatosaurus cervicals (Gilmore, 1936, plate XXIV; Upchurch, Tomida & Barrett, 2005, plate 1) that in this genus the neural spines are proportionally higher relative to the epipophyses in the anterior cervicals than in the posterior. The trend is opposite in Erketu, in which the epipophyses increasingly dominate neural spines anteriorly. This further demonstrates the variety of different mechanical strategies used by different sauropods to support their long necks. In those sauropods without ostensible epipophyses, phylogenetic bracketing nevertheless suggests that muscles did insert above the postzygapophyses, but the insertions are not marked by obvious scars or processes and these muscles were probably less important than those attached to the spine.

Cervical ribs

In extant birds, cervical ribs are the insertion points for the M flexor colli lateralis, M flexor colli medialis and M longus colli ventralis hypaxial muscles (Zweers, Vanden Berge & Koppendraier, 1987; Baumel et al., 1993; see also Fig. 5). No bird has cervical ribs long enough to overlap, but the tendons that insert on the cervical ribs do overlap and are free to slide past each other longitudinally. In less derived saurischians, including sauropods, the ventral tendons are ossified into long, overlapping cervical ribs (Klein, Christian & Sander, 2012) which are secondarily shortened to less than a centrum length in Diplodocoidea and in Maniraptoriformes, including birds. The null hypothesis is that the long cervical ribs of theropods and sauropods functioned similarly to the short cervical ribs and long tendons of birds, as the insertions of long hypaxial muscles. However, some aspects of muscle insertion in sauropod necks are mysterious and may be illuminated by closer comparisons to their extant relatives.

In birds, ossification (or at least mineralization) of tendon has many functional effects: it (1) restricts tendon deformation; (2) reduces tendon strain at a given stress; (3) accommodates higher load bearing (to a point; see below); and (4) reduces damage to the tendon (Landis & Silver, 2002, p. 1153). In general, proportionally longer and thinner tendons are more extensible and allow more elastic recoil, and shorter, thicker tendons are less extensible and provide less elastic recoil (Biewener, 2008, pp. 272–274). Mineralization or ossification reduces the extensibility of a tendon, and can allow a long, thin tendon to behave more like a short, thick one. Ossified tendons in the lower limbs of birds are typically found distal to the knee (Hutchinson, 2002, p. 1071), where the tendons are constrained to be long and thin by the overall construction of the limb; ossification may be the only viable way for birds to advantageously shift the mechanical properties of these tendons.

The long hypaxial tendons in the necks of sauropod dinosaurs may have been similarly constrained. Ossification of the hypaxial tendons into long cervical ribs may have provided several benefits for sauropods: • Long tendons move the bulk of the hypaxial neck muscles closer to the base of the neck, which reduces the lever arm of the neck mass. Tendon has a much lower Young’s modulus than bone, and reducing the elastic recoil of the hypaxial tendons would have allowed the hypaxial muscles of sauropods to more directly affect the vertebrae to which they were attached. Reduced tendon elasticity is known to improve position control of the involved muscles (Alexander, 2002, p. 1009).

• It has been suggested (Wedel, Cifelli & Sanders, 2000b, p. 380) that elongate cervical ribs may have played a role in ventrally stabilizing the neck, i.e., preventing involuntary dorsal extension by contracting antagonistically against the stronger epaxial tension members (which had to counteract gravity in addition to shifting the mass of the neck).

• Stiff cervical ribs would have helped provide lateral stabilization for the neck, which would have been especially important in taxa with epaxial tension members concentrated on the midline (i.e., those with non-bifid spines) as discussed above.

• Stiff cervical ribs would have provided resistance against torsion of the neck.

(It has also been suggested by Martin, Martin-Rolland & Frey (1998) that the cervical ribs of at least some sauropods functioned as incompressible ventral bracing members. But this hypothesis is badly flawed – see Wedel, Cifelli & Sanders, 2000b, p. 379–380; Klein, Christian & Sander, 2012).

If either of the first two hypotheses is accurate, it is difficult to understand why diplodocids evolved apomorphically short cervical ribs, especially long-necked forms such as Barosaurus and Supersaurus. If the primary role of long cervical ribs was in providing lateral stabilization for taxa with midline epaxial tension members, then the need for this stabilization would be reduced in forms with bifid spines, such as diplodocids, which shifted their epaxial tension members laterally as they were attached to the metapophyses. This, however, would raise the question of why other taxa with bifid spines (e.g., Camarasaurus) also retained elongate cervical ribs, and in the case of Mamenchisaurus apparently evolved apomorphically long cervical ribs (Russell & Zheng, 1993, pp. 2089–2090). It may be that these taxa retained their epaxial tension members primarily on the midline, in the intermetapophyseal trough, while diplodocids shifted theirs laterally; but we know from osteological evidence (see above) that at least some diplodocids did have ligaments or muscles anchored within the trough. Mallison (2011, p. 238) suggested that the short cervical ribs of diplodocids could be an adaptation for neck flexibility. This would be useful in tripodal feeding in dense canopies – a behaviour that their hips and hindlimbs were adapted for.

Apatosaurus presents a final riddle regarding cervical ribs. Even among diplodocids, it had extraordinary cervical ribs: very short, very robust, and positioned very low, far below the centra on extremely long parapophyses (Figs. 7.1 and 7.2), so that the neck of Apatosaurus must have been triangular in cross-section. What function can the ribs have evolved to perform? They were much too short to have functioned efficiently in horizontal or vertical stabilization, and in any case seem over-engineered for these functions. It is tempting to infer that the autapomorphies of the neck in Apatosaurus are adaptations for some unique aspect of its lifestyle, perhaps violent intraspecific combat similar to the “necking” of giraffes. Even if this were so, however, it is difficult to see the benefit in Apatosaurus excelsus Marsh, 1879a of cervical ribs held so far below the centrum – an arrangement that seems to make little sense from any mechanical perspective, and may have to be written off as an inexplicable consequence of sexual selection or species recognition.

Asymmetric elongation of cervical ribs and epipophyses

A central paradox of sauropod cervical morphology is that in the elongation of the cervical ribs, the vertebrae appear better adapted for anchoring hypaxial than epaxial musculature – even though holding the neck up was important and, due to gravity, much more difficult than drawing it down. First, the cervical ribs present a greater area for muscle attachment than the epipophyses do; and second, the much greater length of the cervical ribs in most sauropods enabled the hypaxial musculature to be shifted backwards much further than the epaxial musculature, as the epipophyses are not elongate in any known sauropod. We know that posterior elongation of the epipophyses is developmentally possible in saurischians, because those in the tail of Deinonychus Ostrom, 1969a are extended to the length of a centrum (Ostrom, 1969b, figure 37). Figure 10 shows the cervical skeleton of Euhelopus as it actually is, and reconstructed with speculative muscle attachments that would have been more mechanically efficient: why did sauropod necks not evolve this way? In fact, there are several likely reasons.

Figure 10 Real and speculative muscle attachments in sauropod cervical vertebrae.

1, The second through seventeenth cervical vertebrae of Euhelopus zdanskyi Wiman, 1929 cotype specimen PMU R233a-δ (“Exemplar a”). 2, Cervical 14 as it actually exists, with prominent but very short epipophyses and long cervical ribs. 3, Cervical 14 as it would appear with short cervical ribs. The long ventral neck muscles would have to attach close to the centrum. 4, Speculative version of cervical 14 with the epipophyses extended posteriorly as long bony processes. Such processes would allow the bulk of both the dorsal and ventral neck muscles to be located more posteriorly in the neck, but they are not present in any known sauropod or other non-avian dinosaur. Modified from Wiman (1929, plate 3).

• First, positioning and moving the neck for feeding would have required fine control, and precise movements requires short levers.

• Second, although bone is much stiffer than tendon, it is actually not as strong in tension, so that an ossified tendon is more likely to break under load.

• Third, muscles expand transversely when contracted lengthways. For epaxial muscles in sauropods necks, this expansion would strongly bend ossified epipophyseal tendons, subjecting them to greater stress than simple longitudinal tension. (The same effect would also have caused some bending of cervical ribs, but the lower stresses in ventral musculature would have reduced the effect.)

Short neural spines in long necks

In many cases, the sauropods with the proportionally longest necks are also those whose necks superficially make the least mechanical sense. It is particularly notable that mamenchisaurids (Mamenchisaurus and Omeisaurus) have very low neural spines, as does Erketu in the preserved, anterior, cervicals. Since excessively long neural spines would impede neck extension by overlapping with each other, as in dicraeosaurids, shorter spines would be advantageous for improving neck flexibility. But these low spines would have reduced the lever arm with which epaxial tension members acted.

A speculative explanation, at least, can be offered. Counter-intuitively, the height above the centrum at which a muscle of given size acts has no effect at all on its ability to move the vertebra through a given arc. Although muscles attached to a short spine need to exert greater force to allow for the shorter lever arm, they correspondingly need contract a shorter distance in order to raise the neck by the same amount. Low neural spines, then, may have been connected by strongly pennate muscles, able to contract very forcefully but only over a short distance.

In an animal adopting this low-spine strategy to neck elongation, the difficulty is simply one of fitting the muscle into the space available. A lower limit to neural spine length is imposed by the volume of muscle needed to produce the range of motion. (Raising the neck is work, and while the force exerted by a muscle is proportional to its cross-sectional area, the work it can do varies with volume, so shorter muscles need a correspondingly larger cross-sectional area.)

Another possibility is that taxa with short spines had shifted almost all of their epaxial muscle attachments to the epipophyses, as with the long dorsal muscles of birds. In birds, the long multisegment epaxial muscles are free to “bowstring” across the dorsal curvature at the base of the neck (van der Leeuw, Bout & Zweers, 2001, figure 5). Short neural spines do not indicate poor mechanical advantage for these muscles, because they act at high angles of inclination to the long axis of each vertebra. Tall neural spines increase mechanical advantage of muscles when the vertebrae are held horizontally, but this is unlikely to have been a common posture for sauropods (Taylor, Wedel & Naish, 2009).

Homology and analogy of vertebral features

Bony attachment sites for the large cervical muscles have varied along the evolutionary line from basal amniotes to birds (Fig. 11). In extant lizards and crocodilians, as in basal archosaurs (Fig. 11.1), the neural spine is very large and anchors essentially all of the large multisegment epaxial muscles (Tsuihiji, 2005, figure 2), and there are no epipophyses at all. However, in extant birds, as in non-avian theropods (Figs. 11.4, 11.5), the epipophyses are more prominent and significant than the neural spines and serve as insertion points for all of the multisegment dorsal muscles. The very short neural spines serve as the origins of long dorsal muscles running anteriorly from each vertebra, but the only muscles that insert on the spines and adjacent bony ridges are the small and short Mm. interspinales and Mm. intercristales.

Figure 11 Archosaur cervical vertebrae in posterior view, Showing muscle attachment points in phylogenetic context.

Blue arrows indicate epaxial muscles attaching to neural spines, red arrows indicate epaxial muscles attaching to epipophyses, and green arrows indicate hypaxial muscles attaching to cervical ribs. While hypaxial musculature anchors consistently on the cervical ribs, the principle epaxial muscle migrate from the neural spine in crocodilians to the epipophyses in non-avial theropods and modern birds, with either or both sets of muscles being significant in sauropods. 1, Fifth cervical vertebra of Alligator mississippiensis, MCZ 81457, traced from 3D scans by Leon Claessens, courtesy of MCZ. Epipophyses are absent. 2, Eighth cervical vertebra of Giraffatitan brancai paralectotype HMN SII, traced from Janensch (1950, figure 43 and 46). 3, Eleventh cervical vertebra of Camarasaurus supremus, reconstruction within AMNH 5761/X, “cervical series I”, modified from Osborn & Mook (1921, plate LXVII). 4, Fifth cervical vertebra of the abelisaurid theropod Majungasaurus crenatissimus, UA 8678, traced from O’Connor (2007, figures 8 and 20). 5, Seventh cervical vertebra of a turkey, Meleagris gallopavo, traced from photographs by MPT.

In intermediate forms such as sauropods the situation is more complex, as both the neural spines and epipophyses are prominent – to varying degrees in different species. In sauropods with unsplit neural spines, such as Giraffatitan (Fig. 11.2), the muscles of the neural spine were presumably significant, and would have acted primarily along the midline of the neck. The muscles of the epipophyses were also present, but because their insertions are positioned laterally, the action of these muscles would have functioned both in support and in lateral movement. In sauropods with bifid neural spines, such as Camarasaurus (Fig. 11.3), the muscles inserting on the neural spine were also laterally displaced, so that they as well as the Mm. longus colli dorsalis would have had the dual function of support and lateral motion. In sauropods with bifid spines, then, the one- or two-segment Mm. intercristales and Mm. interspinales shared the function of lateral stabilization and movement with the multisegment Mm. longus colli dorsalis.

It is tempting to imagine an evolutionary pathway in which bifurcation of neural spines was an intermediate step in the evolutionary shift of the insertions of the large multisegment epaxial muscles from the neural spine to the epipophyses. However, this explanation cannot be correct, as bifid spines are not known in taxa along the line to birds – only in sauropods and a few modern birds. This is particularly clear in Fig. 11.4, where a small neural spine remains in the cervical of Majungasaurus, but is dominated by epipophyses. The sequence instead seems simply have been one of progressive reduction of the neural spine and enlargement of the epipophyses. The outcome of this evolutionary sequence, as shown in Fig. 11.5, is that from a myological perspective, modern birds have functionally bifid neural spines: that is, their vertebrae have evolved in a way that is analogous with the true bifid spines of sauropods even though it is not homologous.

Conclusions: Why Giraffes Have Such Short Necks

Reviewing the characters that facilitate the evolution of extremely long necks, it is apparent that only sauropods have them all (Table 3). Although the necks of giraffes are the longest of any extant animals, they are shorter by a factor of six than those of the longest sauropods, because giraffes have relatively small torsos, relatively large, heavy heads, only seven cervical vertebrae, no air-sac system and no vertebral pneumaticity. Absence of elongated cervical ribs may also impede neck elongation. In defence of giraffes, they are relative latecomers in evolutionary terms: given a few tens of millions more years, it is conceivable that they might overcome some of these disadvantages to evolve longer necks. But in some respects they seem locked into a mammalian pattern that will always prevent them from matching the necks of sauropods: extensive oral processing of food requires a large head with heavy teeth; almost no mammal has evolved more than seven cervical vertebrae; and the mammalian lung has attained a local maximum of efficiency that makes it unlikely ever to evolve into something analogous to the avian flow-through lung, so both an air-sac system and vertebral pneumaticity are precluded.

Table 3 Neck-elongation features by taxon.

	Absolutely large body size	Quadrupedal stance	Small head	Numerous cervical vertebrae	Elongate cervical vertebrae	Air-sac system	Vertebral pneumaticity	
Human								
Giraffe		✓			✓			
Ostrich			✓	✓	✓	✓	✓	
Paraceratherium	✓	✓						
Therizinosaurus	✓		✓			✓	✓	
Gigantoraptor	✓		✓			✓	✓	
Arambourgiania					✓	✓	✓	
Sauropods	✓	✓	✓	✓	✓	✓	✓	

Similarly, ostriches seem unlikely ever to evolve really long necks, despite the prerequisite small heads and avian lung, simply because they are small bipeds. Birds seem unable to attain sizes exceeding the 500 kg of the “elephant bird” Aepyornis maximus, probably because adult body size and egg size, which were not tightly correlated in non-avian dinosaurs, are correlated in birds (Birchard & Deeming, 2009; Deeming & Birchard, 2009). There are strong mechanical constraints on the latter: as body size increases, the eggs approach a point at which the shell cannot simultaneously be thick enough to support the egg and thin enough for the hatchling to break out of (Murray & Vickers-Rich, 2004, p. 212; Birchard & Deeming, 2009).

Some of the other long-necked taxa listed in Table 3 seem to have been better equipped to evolve longer necks. It is impressive that the azhdarchid pterosaurs seem likely to have achieved 3m while retaining flight: no doubt their pneumaticity was a key feature in making this possible in spite of their large heads. Nevertheless, the absolute size constraints imposed by flight make it unlikely that pterosaurs would have greatly exceeded this mark even had they survived the end-Cretaceous extinction.

The two theropod clades mentioned above (therizinosaurs and to a lesser extent oviraptorosaurs) appear to have had small heads, proportionally similar in size to those of sauropods, as well pneumatic systems that invaded their vertebrae. Why did they not evolve necks as long as those of sauropods? Possible reasons include the following: • All theropods were bipedal, and the demands of bipedal locomotion may have prevented them from evolving the giant body sizes that are required for very long necks.

• The long-necked theropods may not have been under the same selection pressure to evolve long necks as were sauropods. If they were omnivorous, for example, then their use of more nutritious food may have mitigated the need for increased feeding envelopes. Among extant theropods, the ostrich is proportionally long-necked but feeds mostly from the ground (Dzemski & Christian, 2007), and so has no selective pressure to evolve a yet longer neck.

• All of the largest long-necked theropods lived in the Late Cretaceous, two of them in the Campanian–Maastrichtian. Had they not died out at the end of the Cretaceous, they might have gone on to attain larger size. On the other hand, sauropods attained large size very quickly in evolutionary terms, with a 104 cm humerus from the late Norian or Rhaetian indicating a Camarasaurus-sized sauropod only about ten million years after the first known dinosaurs (Buffetaut et al., 2002). If theropods did not evolve larger body size in the 150 million years available to them, it seems likely that they did not have the potential to do so.

• Finally, it should be noted that both of the long-necked theropods discussed above are known from incomplete remains that do not include any informative cervical material. It is possible that neck length was positively allometric in these clades, as in sauropods, and they may have had necks somewhat longer than isometric scaling suggests.

In summary, no other clade has all of the suggested adaptations for long necks that are found in sauropods. Were it not for the end-Cretaceous extinction, non-avian theropods would have been the most likely candidates for evolving sauropod-like neck lengths, due to the combination of pneumaticity, small heads in some clades. and potential for large body size. Museum Abbreviations

BYU Earth Sciences Museum, Brigham Young University, Provo, Utah (USA)

CM Carnegie Museum of Natural History, Pittsburgh, Pennsylvania (USA)

MfN Museum für Naturkunde Berlin, Berlin (Germany) (collection numbers for fossil reptiles: MB.R.####)

FMNH Field Museum of Natural History, Chicago, Illinois (USA)

IGM Geological Institute of the Mongolian Academy of Sciences, Ulaan Baatar (Mongolia)

ISI Geology Museum, Indian Statistical Institute, Calcutta (India)

MAL Malawi Department of Antiquities Collection, Lilongwe and Nguludi (Malawi)

MCZ Museum of Comparative Zoology, Harvard University, Cambridge, Massachusetts (USA)

MIWG Dinosaur Isle, Sandown, Isle of Wight (UK)

OMNH Oklahoma Museum of Natural History, Norman, Oklahoma (USA)

PIMUZ University of Zurich Paleontology Museum, Zurich (Switzerland)

PMU Palaeontological Museum, Uppsala (Sweden)

UA Université d’Antananarivo, Antananarivo (Madagascar)

UJF University of Jordan Department of Geology Collections, Amman (Jordan)

WDC Wyoming Dinosaur Center, Thermopolis, Wyoming (USA)

ZMNH Zhejiang Museum of Natural History, Hangzhou (China)

We thank RL Cifelli, NJ Czaplewski, and J Person (Oklahoma Museum of Natural History), DS Berman, MC Lamanna, and AC Henrici (Carnegie Museum of Natural History), BB Britt, KL Stadtman, and RD Scheetz (Brigham Young University), LL Jacobs and DA Winkler (Southern Methodist University), and S Hutt (Dinosaur Isle) for access to specimens. DT Ksepka (American Museum of Natural History) provided high-resolution versions of the figures from his description of Erketu and LPAM Claessens (College of the Holy Cross) provided unpublished images of alligator vertebrae. DM Lovelace (Wyoming Dinosaur Center) provided a cross-sectional photo of a broken Supersaurus cervical for the ASP calculations. DWE Hone investigated the status of the Omeisaurus junghsiensis material and allowed us to note his conclusion. MP Witton (University of Portsmouth) provided helpful discussion on pterosaur necks. WI Sellers (University of Manchester) clarified our understanding of mechanical advantage. We used translations of several papers from the Polyglot Paleontologist web-site (http://www.paleoglot.org/index.cfm). We thank Scott Hartman for kindly allowing us to use several of his skeletal reconstructions in parts of Figs. 1 and 2; and “kei” and Kevin Ryder for permission to use their photographs. This paper has been reviewed by at least eight referees at various times. We thank those who provided useful feedback and helpful contributions – in particular H Mallison (Museum für Naturkunde Berlin) and the anonymous referee who also reviewed this version. Finally we thank J Hutchinson (Royal Veterinary College) for his constructive and efficient editorial handling of the manuscript.

Additional Information and Declarations

Competing Interests

Author Contributions

No competing interests.

Michael P. Taylor and Mathew J. Wedel conceived and designed the experiments, performed the experiments, analyzed the data, wrote the paper.

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
