# Peer review of "Why sauropods had long necks; and why giraffes have short necks"

_PeerJ, doi:10.7717/peerj.36_

## Round 0.1 · original submission · Minor Revisions

The reviewers note that the paper is very review-like but with enough insight to merit eventual publication (one could call it a "meta-analysis"). I will email the marked-up pdf from Mallison. I agree that if it is to make broadly applicable statements about long necks it should cast a wider net, especially within Dinosauria. It definitely should be made clearer that it is pretty well accepted that mammalian necks have some sort of genetic/developmental constraint on vertebral number (except a few oddballs), including giraffes, as per the 2nd reviewer.

Based on the revisions detailed in your Rebuttal, I'll decide if the paper needs re-review- a serious, rigorous effort would mean no need for re-review. Consider this provisional acceptance with the caveat that moderate revisions are needed as per the reviewers' comments.

·

Basic reporting

Minor points:
- sometimes, the authors go a bit too far trying to write in easily understandable, conversational tone. Clarity can suffer from bad language and from oversimplification, the balancing act doesn't work sometimes. I have marked these instances in the PDF.
- the structure is unusual, but helps readability a lot.

Experimental design

This is a review article, thus it relies exclusively on previously published data.
The review is not universal and all-encompassing, and no claim of completeness is made. The authors should consider including some of the better-known flukes such as Tanystropheus (proportionally long neck, very elongate vertebrae, ecology of neck totally unclear), even if all they can say is that science is clueless.

Validity of the findings

The authors conflate proportionally long necks and absolutely long necks. This needs to be fixed by being more specific at each mention, and be shortly introducing the difference.
Furthermore, the authors gallantly omit non-sauropod Sauropodomorpha and basal Dinosauria. However, basal dinosaurs had proportionally long necks, and "prosauropods" start all the key features that allowed extremely long necks (proportionally and absolutely) in sauropods. They should expand and adapt their manuscript to detail the early history of long necks more.
Lastly, the difference between a long neck and an extremely long neck is of importance (and ties in with absolute size), and should be discussed. Maybe the author can define a term for the longer-than-proportional (even for sauropodomorphs) necks that are the focus of the paper?

Additional comments

I chose "major revisions" because this smack in between minor and major. A re-review may not be necessary, depending on how much of my suggestions you follow. Otherwise, please see the edited PDF.

Reviewer 2 ·

Basic reporting

p. 6 - anterior limb elongation has a lot to do with supporting the long neck - also tail - so expand more on this.

p. 8 reaches is apotheosis - remove the is BUT isn't apotheosis a bit too much to use the term in anatomy......?

p. 8 - if you avoid plesiosaurs then why you cover whales in detail? I think you should consider the plesiosaurs despite their aquatic modes. I like the whale part.

the bifid spinous processes (also in humans) are due to bilateral symmetry. the same for the ligs and muscles. in theory there is just fascia at the median plane. Now if bilateral structures do not show, this implies compression and fusion at the median plane.

figure 8 re-draw and label - it is confusing to see the extensions

the part with the ostrich neck needs more in depth coverage. I am not convinced that the more muscular neck option is totally unlikely. Compare a small bid muscle bulk of neck with the ostrich to see how size affects the situation in birds.

Experimental design

it is fine as it is impressively detailed coverage - the descriptions are fine

Validity of the findings

The data are fine.

Well -- a lot of it is guessing and second order assumptions but they are the only thing to be done - it is paleontology - so it is necessary to do this - we cannot do experiments with fossils. the comparative extant situation is sound and supportive.

Additional comments

I liked the article - I found the giraffe situation unnecessary but if you like to see it in this prism it is OK - I see no new information on the giraffe aspect and no one expected more vertebrae in the giraffes - we know mammals are genetically locked.

---

## Round 0.2 · Minor Revisions

Apologies for the delay in the decision, which was due to a delayed informal clarification from a reviewer. However, we are pleased to report that there are just some very minor corrections to be made and then we can accept the paper. Reviewer 2 essentially accepted the rebuttal, and I deem your revisions to Reviewer 1 to be adequate and not requiring re-review. However, these final changes came up:

The speculations in the MS on Deinocheirus's neck length are unnecessary and excessive; they could lead to confusion and misinformation in the literature. Scaling fossils consisting of just forelimbs to whole body reconstructions and neck length data is not good practice in paleontology. It is not even certain (yet) that Deinocheirus is an ornithomimosaur; some recent studies have questioned this. Better to leave it out entirely from the paper. Gigantoraptor is based on more published actual data and fills the same need of having a large-ish theropod represented; same as for Therizinosaurus and Tyrannosaurus. With those 3 taxa you are well covered for theropods.

Sorry for noticing this late-- I wish I'd spotted it earlier, but I feel this is an important amendment to make, otherwise I would let it go. The text and figures need to be changed to accomodate this. As soon as that is done, I will happily accept the paper.

---

## Round 0.3 · accepted · Accept

Well done! I am happy with the MS as it stands, so as I said I can happily accept it.